# Multi-Scale Depthwise Separable Capsule Network for hyperspectral image classification

Lin Wei[1,2], Haoxiang Ran[1]*, Yuping Yin[3], Huihan Yang[4]

1 School of Software, Liaoning Technical University, Huludao, Liaoning, China, 2 The Department of Basic Education, Liaoning Technical University, Huludao, Liaoning, China, 3 Faculty of Electrical and Control Engineering, Liaoning Technical University, Huludao, Liaoning, China, 4 School of Electronic and Information Engineering, Liaoning Technical University, Huludao, Liaoning, China

☯ These authors contributed equally to this work.
* 3295906115@qq.com

**Data Availability Statement:** The study selected three publicly available hyperspectral remote sensing imaging datasets for ablation and comparative experiments. These include: the Kennedy Space Center dataset, available from

## Abstract

Addressing the challenges in effectively extracting multi-scale features and preserving pose information during hyperspectral image (HSI) classification, a Multi-Scale Depthwise Separable Capsule Network (MDSC-Net) is proposed in this article for HSI classification. Initially, hierarchical features are extracted by MDSC-Net through the employment of parallel multi-scale convolutional kernels, while computational complexity is reduced via depthwise separable convolutions, thus reducing the overall computational load and achieving efficient feature extraction. Subsequently, to enhance the translational invariance of features and reduce the loss of pose information, features of various scales are processed in parallel by independent capsule networks, with improvements in max pooling achieved through dynamic routing. Lastly, features of different scales are concatenated and integrated through the concatenate operation, thereby facilitating precise analysis of multi-level information in the hyperspectral image classification process. Experimental comparisons demonstrate that MDSC-Net achieves average accuracies of 94%, 98%, and 99% on the Kennedy Space Center, University of Pavia, and Salinas datasets, respectively, indicating a significant performance advantage over recent HSI classification models and validating the effectiveness of the proposed model.

## 1 Introduction

Hyperspectral imaging (HSI), as a remote sensing data source rich in physical and chemical properties, has demonstrated immense potential for applications in fields such as earth science, resource exploration, and environmental monitoring [1–3]. In these applications, the task of HSI classification fundamentally involves the precise identification of land cover based on the spectral characteristics of pixels, thereby aiding scientific research and decision-making. The classification of HSI, especially its feature extraction, occupies a crucial position in the domains of machine learning and image analysis, and has long been a subject of keen interest among

https://ehu.eus/ccwintco/index.php?title=
Hyperspectral_Remote_Sensing_
Scenes#Kennedy_Space_Center_(KSC); the
University of Pavia dataset, accessible via https://
ehu.eus/ccwintco/index.php?title=Hyperspectral_
Remote_Sensing_Scenes#Pavia_Centre_and_
University; and the Salinas dataset, which can be
found at https://ehu.eus/ccwintco/index.php?title=
Hyperspectral_Remote_Sensing_Scenes#Salinas.

**Funding:** This work was supported in part by the
Basic Scientific Research Projects of Education
Department of Liaoning Province under Grant
LJKMZ20220687, "Hyperspectral Image
Classification Based on Multi-layer Deep Learning
Method with Small Samples". It was also supported
by the Liaoning Provincial Natural Science
Foundation under Grant 1704681991881,
"Research on High-precision Classification of
Hyperspectral Images Using Lightweight Feature
Fusion Based on Self-organizing Multi-layer
Heterogeneous Deep Learning Method".
Furthermore, the research received funding from
the Science and Technology Plan Project of
Huludao City under Grant 2023JH (1) 4/04b,
"Research on High-precision Classification of
Hyperspectral Images Based on multi-layer
attention and puzzle networks". The funders had no
role in study design, data collection and analysis,
decision to publish, or preparation of the
manuscript.

**Competing interests:** The authors have declared
that no competing interests exist.

researchers. Although this task poses significant challenges for machines, precise labels can be created through manual annotation of images, providing high-quality training data for supervised learning and thereby effectively supporting the learning process. I In the field of feature extraction, the main approaches to addressing the problem are divided into two categories: one is based on pixels, and the other is based on regions. In pixel-based classification methods, each pixel is classified individually, primarily based on its independent spectral information, without consideration of its spatial relationship with surrounding pixels [4]. Although straightforward, this approach has clear limitations as it neglects the potential spatial relationships between pixels. Conversely, a more efficient method involves analyzing the data cube as a whole, thereby leveraging both spatial and spectral information [5]. This approach more comprehensively considers the spatial connections between pixels, offering possibilities for more accurate image analysis. Hence, this study focuses on region-based analysis methods.

Neural networks, particularly Convolutional Neural Networks (CNN) that have been developed since the 1970s, have shown significant effectiveness in addressing such issues [6]. Architectures like AlexNet [7], ResNet [8], and GoogLeNet [9], although originally designed for processing the red, green, and blue (RGB) bands in visible light, are equally applicable and effective for multi-channel data analysis. With the widespread application of CNN in image processing, their outstanding performance in HSI classification tasks has been confirmed by numerous studies [10–12]. For instance, 3D-CNN have been used for feature extraction from HSI [13]; the introduction of attention mechanisms [14] has optimized feature map extraction; and network structures like HybridSN [15] and SpectralNET [16] have achieved breakthroughs in HSI classification accuracy. Notably, recent studies [17, 18] have further proposed various methods to enhance HSI classification accuracy. The aforementioned research, although achieving significant accomplishments in HSI classification tasks, still faces limitations in aspects such as the reduction of spatial information due to pooling operations, and large model parameter size. Notably, CNN struggle to effectively recognize object pose information, as their convolutional filters cannot represent feature transformation activities. To overcome the limitations of traditional CNN approaches, the concept of capsule networks was first introduced by Hinton and colleagues [19]. Capsules are groups of neurons that describe an entity's pose and probability of existence, containing more information about the entity's attributes than scalar neurons in CNN. The capsule network (CapsNet), proposed by Sabour and others [20], encodes the probability of an object's presence and its pose through the length and orientation of activity vectors, enhancing the performance of capsule networks in image analysis. Further, H. Zhang and colleagues applied capsule networks to HSI classification [21], achieving significant results. Additionally, Hinton and colleagues proposed matrix capsules with EM routing [22], addressing some deficiencies of the dynamic routing algorithm in [20], such as using the negative log variance of Gaussian clusters to measure the consistency between pose vectors, and representing poses with matrices instead of vector lengths.

Recently, the trend has been to apply capsule network technology to HSI analysis. Methods based on capsule networks have demonstrated significant effectiveness in extracting deep semantic features of hyperspectral images. This approach utilizes neurons within capsule networks to simultaneously capture both the spectral and spatial features of HSI images, as evidenced by several studies [23–25]. Although existing capsule networks have improved some aspects of CNN in HSI classification tasks, they face significant challenges in parsing complex spectral-spatial features that vary with scale. Particularly in capturing and analyzing multiscale features, capsule networks often fail to fully extract spectral characteristics, thereby affecting the effective interpretation of rich geophysical details and multi-level information in HSI, which in turn reduces the accuracy of the classification process. In summary, the problem this study aims to address in HSI classification tasks is that the pooling operations in convolutional

networks often lead to a reduction in spatial information, thereby diminishing the ability to effectively recognize object pose information [26]. Additionally, capsule networks have not been sufficiently capable of extracting multi-scale spectral features [27]. To overcome the limitations of existing methods, we propose an innovative Multi-Scale Depthwise Separable Capsule Network (MDSC-Net). In summary, the main contributions of this paper are as follows:

1. A novel multi-scale capsule network architecture is introduced, utilizing multi-scale convolutional kernels to extract features at different levels, effectively parsing the rich detail and multi-layered information in hyperspectral images. This enhances the precise identification and understanding of complex terrains and land cover types. Feature maps at each scale are processed independently by capsule networks to accurately capture subtle changes in land cover at different scales, ensuring the integrity of spatial information and improving the efficiency of hyperspectral image data analysis.

2. By employing a dynamic routing mechanism instead of the traditional max-pooling method, the dimensionality of feature maps is effectively reduced, and the pose matrices in capsule networks enhance the model's translation invariance of features. This approach improves the network's ability to capture details when handling complex terrains and ensures the in-depth analysis of the rich hierarchical information in hyperspectral image data, achieving highly accurate and efficient data processing.

3. Depthwise separable convolution is incorporated into the architecture by replacing the original three-dimensional convolution structure with depthwise and pointwise convolutions. This ensures efficient feature extraction while significantly reducing the model's computational complexity, thereby effectively alleviating the overall computational burden.

3. In addition to evaluating the effectiveness of the proposed MDSC-Net architecture in terms of overall accuracy (OA), average accuracy (AA), and kappa coefficient (Kappa), this method is also compared with existing state-of-the-art hyperspectral image classification methods in terms of efficiency.

## 2 Basic principles of the EM routing algorithm

The EM routing algorithm plays a crucial role in selecting the optimal capsule pathways to optimize information flow within this model. The algorithm involves M-steps and E-steps. The E-step equation is as follows:

$$Q(\theta|\theta^t) = E[\log L(\theta|Z)|X, \theta^t] \tag{1}$$

Where $\theta$ represents parameters, $Z$ is the latent variable, $X$ is the observed variable, $L$ is the observed variable, $t$ denotes the iteration number. This step selects the optimal output capsule by evaluating the response of each capsule to the observed data. In the M-step, our objective is to maximize the $Q$ function. The M-step equation is as follows:

$$\theta^{t+1} = \arg\max_\theta Q(\theta|\theta^t) \tag{2}$$

The $Q$ function is defined as the expectation $\theta^t$ of the complete data log-likelihood function $L$ of the observed data $X$ with respect to $Z$, given the current parameter estimates. This expectation is used to update the model parameters in each iteration of the EM algorithm. The convergence property of the EM algorithm ensures that $\log L(\theta^t|X)$ does not decrease after each iteration, ultimately converging to a local maximum or saddle point.

To better understand the process, here is a specific example:

Assume we have a simple binary classification capsule network where each input vector can belong to two different capsule output classes. The initial parameter settings are $\theta^{(0)}$, which includes the means $\mu_1^{(0)}$ and $\mu_2^{(0)}$, and the variances $\sigma_1^{(0)}$ and $\sigma_2^{(0)}$ for the two classes. The E-step

uses Bayes' theorem to calculate the posterior probability $x_i$ of each observed data point $\gamma$ belonging to each class. In the $t$-th iteration, we compute the probability of each input vector $x_i$ belonging to the two output capsules. Given the observed data $X = \{x_1, x_2, \ldots, x_N\}$ and the latent variables $Z = \{z_1, z_2, \ldots, z_N\}$, the posterior probability is calculated using Bayes' theorem as follows:

$$\gamma_{i1}^{(t)} = P(z_i = 1 | x_i, \theta^{(t)}) = \frac{\pi_1^{(t)} \mathcal{N}(x_i | \mu_1^{(t)}, \sigma_1^{(t)})}{\pi_1^{(t)} \mathcal{N}(x_i | \mu_1^{(t)}, \sigma_1^{(t)}) + \pi_2^{(t)} \mathcal{N}(x_i | \mu_2^{(t)}, \sigma_2^{(t)})} \tag{3}$$

In this context, $N$ represents a Gaussian distribution, $\pi$ denotes the mixing coefficient, and $\gamma_{i1}^{(t)}$ indicates the probability that, at the $t$-th iteration, the $i$-th sample $x_i$ belongs to the first output capsule. The M-step uses the posterior probabilities calculated in the E-step to update the parameters. The formulas for updating the mean and variance are as follows:

$$\mu_1^{(t+1)} = \frac{\sum_{i=1}^{N} \gamma_{i1}^{(t)} x_i}{\sum_{i=1}^{N} \gamma_{i1}^{(t)}} \tag{4}$$

$$\sigma_1^{(t+1)} = \frac{\sum_{i=1}^{N} \gamma_{i1}^{(t)} (x_i - \mu_1^{(t+1)})^2}{\sum_{i=1}^{N} \gamma_{i1}^{(t)}} \tag{5}$$

By iteratively calculating the posterior probabilities and updating the parameters in this manner, the EM algorithm continuously optimizes the model parameters. Ultimately, it converges to a local maximum or saddle point, thereby selecting the optimal capsule path to enhance information flow.

## 3 Model topology

In this section, the design of the Multi-Scale Deep Separable Network (MDSC-Net) model is presented. Its overall structure is depicted in Fig 1. The architecture of the model is divided

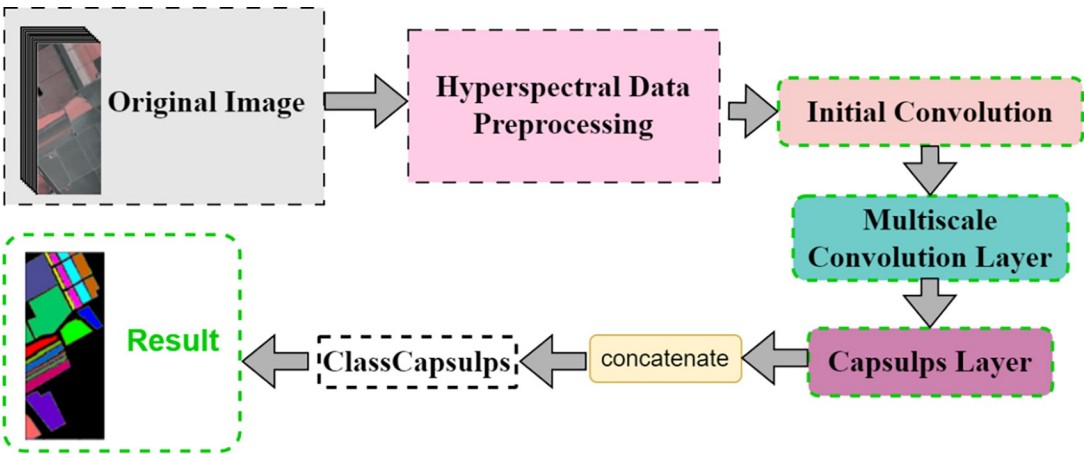

**Fig 1. The topology of this model.**

into six parts, each responsible for different processing and analytical tasks. The first part is dedicated to data preprocessing, where dimensionality reduction is conducted through Principal Component Analysis (PCA), followed by edge padding and image segmentation. In the second part, the initialization layer employs a 5×5 convolutional kernel to preliminarily extract features, providing a foundation for deep learning. The third part utilizes a multi-scale convolutional structure akin to the Inception module, extracting rich features through convolutional kernels ranging from 3×3 to 11×11, thereby enhancing the model's ability to recognize details. The fourth part consists of a capsule layer, including a primary capsule layer and two intermediate al capsule layers. In the fifth part, an integration layer is employed, where the extracted multi-scale features are concatenated. Finally, the classification is performed in the classification capsule layer, where image classification is determined based on the activation values in the feature maps.

### 3.1 Data preprocessing module

The preprocessing stage of the data, as illustrated in Fig 2, aims to enhance computational efficiency, reduce data redundancy, and adjust the data to meet the input requirements of the model.

The original hyperspectral dataset possesses a spatial resolution of $m \times n$ and consists of $k$ spectral bands. Initially, PCA is performed to reduce the dimensionality of the dataset, compressing it from $k$ spectral bands to $d$ bands that contain the most information. This not only improves computational efficiency and reduces data redundancy but also mitigates the impact of noise by discarding those bands that contain less information, as these bands are more susceptible to noise interference. Subsequently, to adapt to the model's input requirements and preserve information at the edges, zero-padding is applied to the dataset. Specifically, considering a selected window size $w$, a zero boundary with a width of p pixels is added around all four sides of the dataset, altering its dimensions to $(m+2p) \times (m+2p) \times d$.

Next, a pixel-wise sliding window operation is performed on the padded image using $w \times w$ pixel blocks (patches), with a stride of 1, ensuring every original pixel can serve as the central point. This process generates $m \times n$ samples, reflecting the region-based approach of the model. Given the prevalence of non-informative background information in the dataset, which is not conducive to the training of the model, samples identified with a background label (label value 0) have been systematically eliminated.

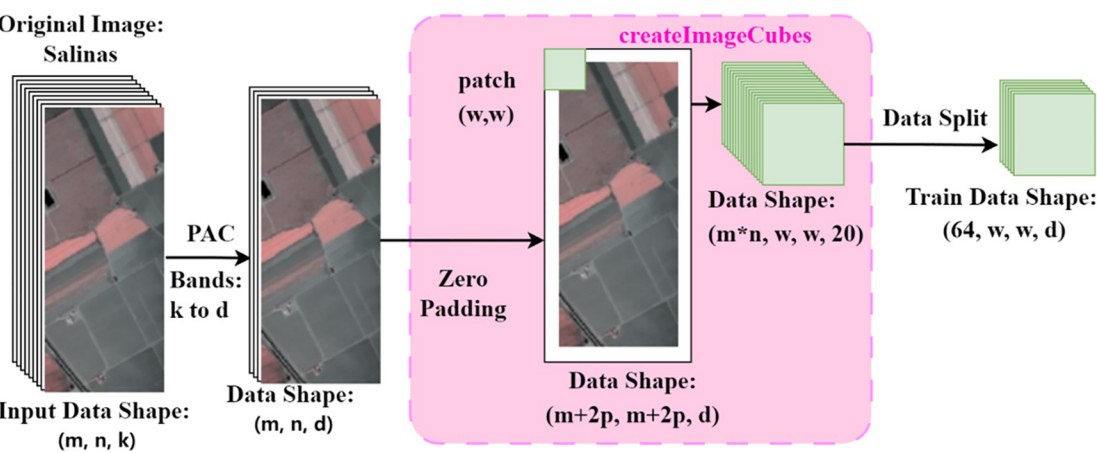

**Fig 2. Data preprocessing.**

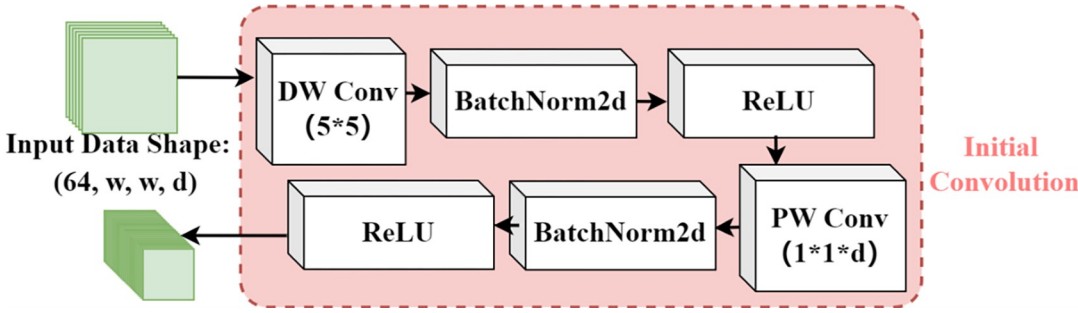

**Fig 3. Initialization convolution module.**

In each training iteration, 64 samples are sequentially selected from the dataset to form a batch. Following the aforementioned preprocessing steps, the input data format for the model is finalized as 64×$w$×$w$×$d$.

### 3.2 Initial convolution and activation module

During the initialization phase of the model, as shown in Fig 3, the aim is to enhance the model's ability to learn and extract complex features from the raw data. Specifically, the first deep convolutional layer (Deep Convolutional Convolution, DW Conv) employs d 5×5 convolutional kernels to extract spatial features from each independent channel.

Subsequently, the data processed by the convolutional layer are subjected to a normalization layer and a Rectified Linear Unit (ReLU) activation layer. The normalization layer standardizes the convolved data, enhancing the model's generalization capability and accelerating the training process. It also aids in reducing noise interference by ensuring that all features are scaled equally, mitigating the unequal impact of noise on features at different scales. The ReLU layer introduces non-linearity, thereby boosting the model's ability to learn complex data structures. After processing through these two layers, the structural dimensions of the data remain unchanged.

The pointwise convolution layer (PW Conv), using 64 1×1×$d$ convolutional kernels, follows next. It integrates the outputs from the normalization and activation layers and adjusts the number of output channels. Another pass through a normalization layer and a ReLU activation layer completes the feature extraction function of the initialization convolution module on the raw data. In this manner, 64 distinct feature maps are extracted from the channels of the raw data, with two example feature maps illustrated in Fig 4.

### 3.3 Multi-scale capsule convolution module

The multi-scale capsule convolutional module, illustrated in Fig 5, is designed to extract multiscale features and pose information from HSI, thereby enhancing the model's performance and classification accuracy on complex datasets. The module employs a structure akin to the Inception model for processing data activated by ReLU. The initial stage includes an adjustment layer with 1×1×64 convolutional kernels, aimed at modulating the number of feature maps for subsequent multi-scale convolutional layers.

Specifically, for the paths processing 3×3 and 5×5 convolutional kernels, the output of this adjustment layer comprises 96 feature maps. The subsequent 3×3 and 5×5 depth convolutional layers further process these feature maps, ultimately outputting 128 feature maps through a 1×1×96 pointwise convolutional layer. This design is intended to capture more refined spatial

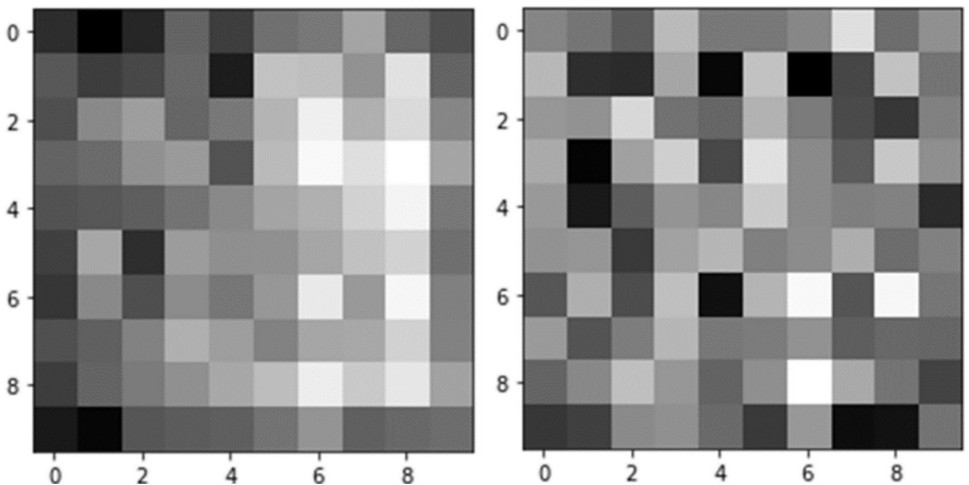

**Fig 4. Initial convolution feature maps.**

features while enhancing the feature extraction capability by increasing the number of convolutional kernels.

On the other hand, for paths with larger convolutional kernels (i.e., 7×7, 9×9, 11×11), the adjustment layer outputs 16 feature maps. After processing by 7×7, 9×9, 11×11 depth convolutional layers and a 1×1×16 pointwise convolutional layer, each layer ultimately outputs 64 feature maps. This approach not only reduces the number of parameters but also effectively captures coarse spatial features by adjusting the use of convolutional kernels of different sizes, thereby achieving a balance between model simplification and precise feature extraction in HSI processing.

Following these convolutional layers is a primary capsule layer that integrates pose information and activation values. Subsequently, the model incorporates two intermediate capsule layers. Ultimately, the feature maps output by the five different convolutional kernel pathways are concatenated along the feature map dimension. At the end of the model, classification is performed by a classification capsule layer. Unlike traditional fully connected layers, this classification layer uses the activation values within the feature maps to vote, facilitating effective classification.

### 3.4 Capsule layer module

The capsule layer structure within the multi-scale capsule convolution module is illustrated in Fig 6. The Primary Capsule Layer enhances the richness and precision of feature representation by integrating pose information and activation values. In this layer, the calculation formula for pose information is $B×P^2$, where $B$ represents the number of capsule types and $P$ denotes the dimensions of the pose matrix. Using a 1×1×128 convolutional kernel, the pose layer generates $B×P$ feature maps containing pose information. Similarly, the activation layer employs a 1×1×128 convolutional kernel to produce 32 feature maps containing activation values, culminating in a feature map dimension of $B×P+32$. Subsequently, the model incorporates two intermediate capsule layers, both utilizing the Expectation-Maximization (EM) routing algorithm for optimized processing. Specifically, the first intermediate capsule layer comprises a 3×3 depth convolutional layer and a 1×1×544 pointwise convolutional layer, aimed at capturing more complex feature information. The second intermediate capsule layer includes a 3×3 depth convolutional layer and a 1×1×272 pointwise convolutional layer, further refining feature extraction and processing.

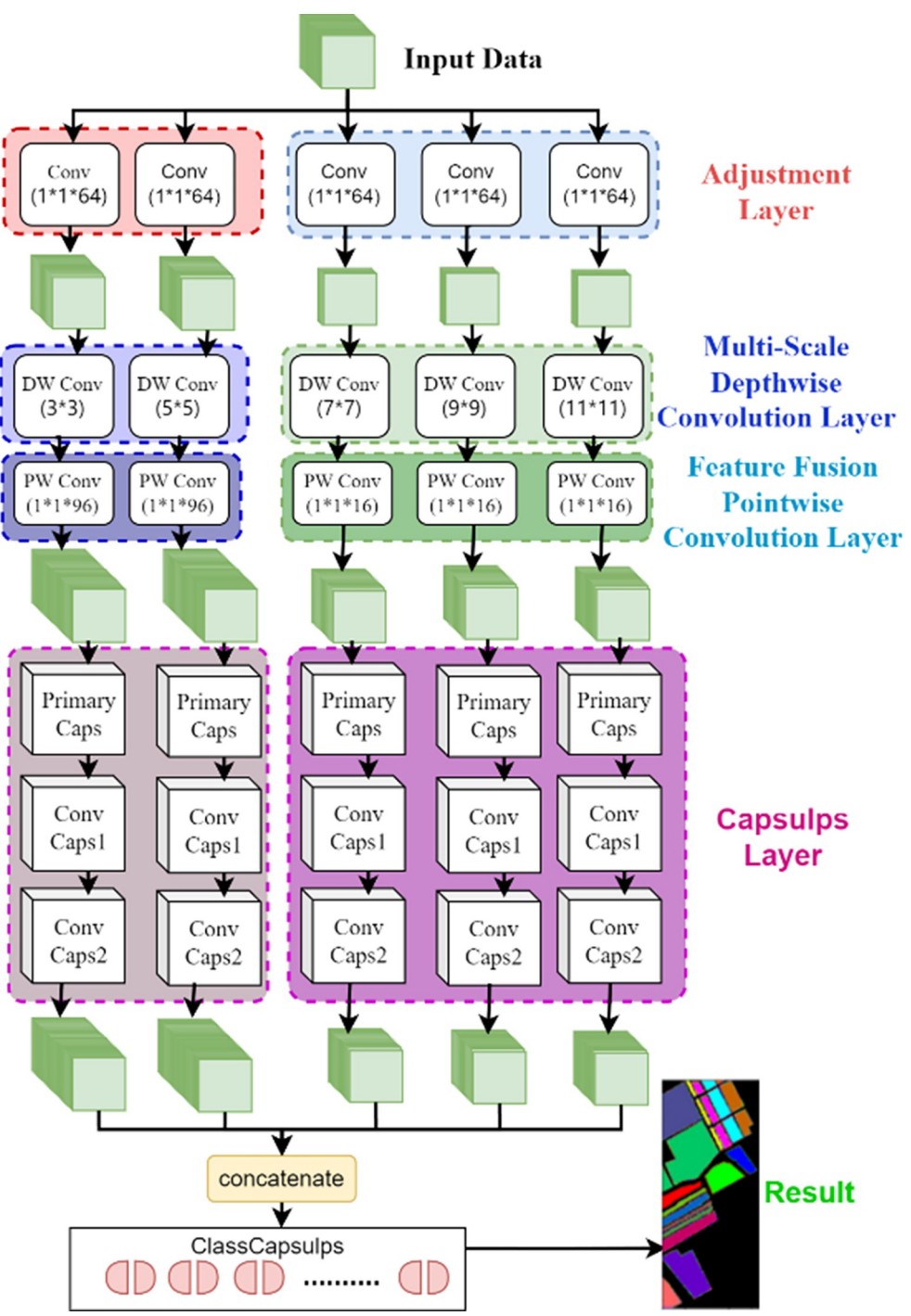

**Fig 5. Multi-scale capsule convolution module.**

### 3.5 Loss function

In capsule networks, the transformation matrix $W$ converts the outputs of lower-level capsules (a group of neurons) into the inputs for higher-level capsules, allowing the lower-level capsules to predict the activation states of the higher-level capsules. The MDSC-Net employs a spread

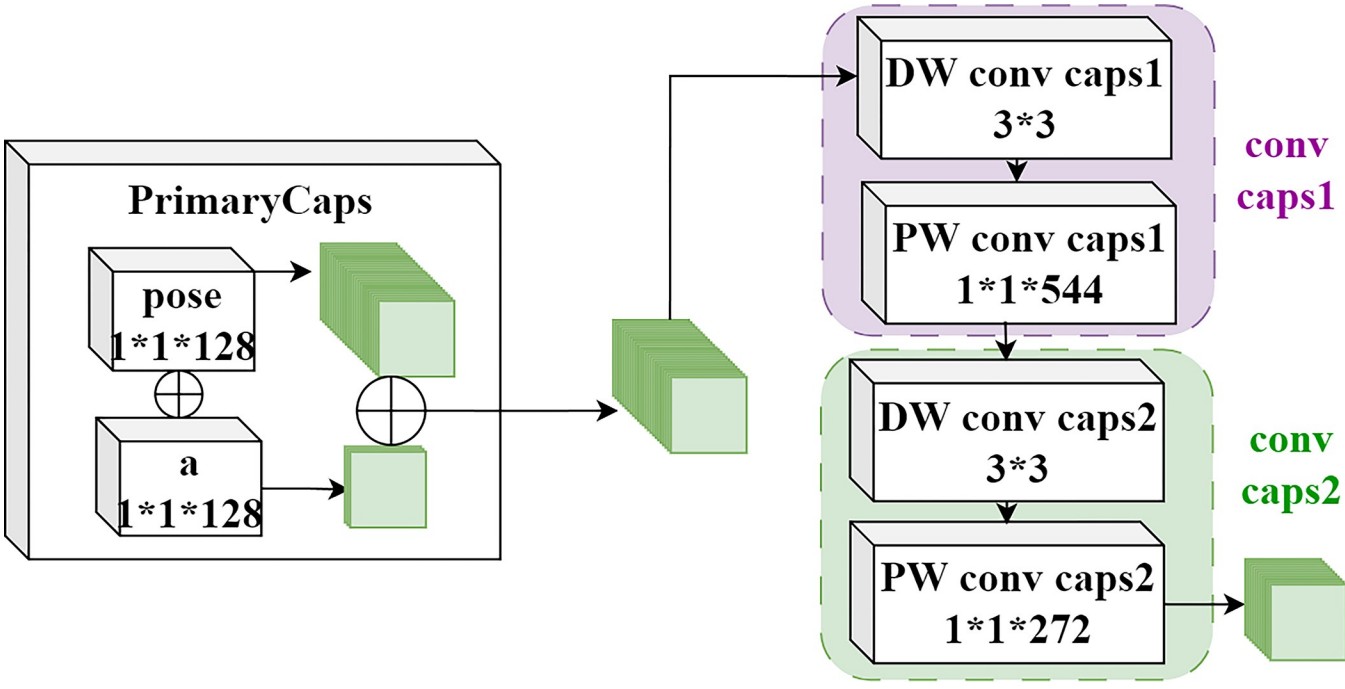

**Fig 6. Capsule layer module.**

loss function for training this transformation matrix *W*. This loss function is specifically designed for optimizing capsule networks, aiming to diminish the impact of model weight initialization and to streamline the hyperparameter tuning process. The calculation of Li's spread loss is as follows:

$$Li = (max(0, m - (at - ai)))^2 \tag{6}$$

Here, *m* represents the margin, while *at* and *ai* denote the activations of the target class *t* and class *i*, respectively. The model incurs a penalty for the squared difference between predictions and the margin *m* when the disparity between the true class and other classes is less than *m*. To prevent the emergence of ineffective capsules at the onset of training, the initial margin *m* is set to a lower value of 0.2.

### 3.6 Model architecture

The detailed explanation of the MDSC-Net attention-dense network model includes the types of layers used, the dimensions of the generated output maps, and the required number of parameters. The specific layer types and their corresponding parameters are illustrated in detail in Fig 7.

## 4 Experimental results and analysis

### 4.1 Experimental setup and dataset description

All the expriments and testing environment were carried out on the PyTorch 1.10.2 framework, accelerated with CUDA 11.3. The Adam optimizer was selected to combine the exponential decay learning rate to optimiza the strategy training model. The initial learning rate was set at 0.003 and decreased with a decay rate of 0.96 as the number of iterations increased.

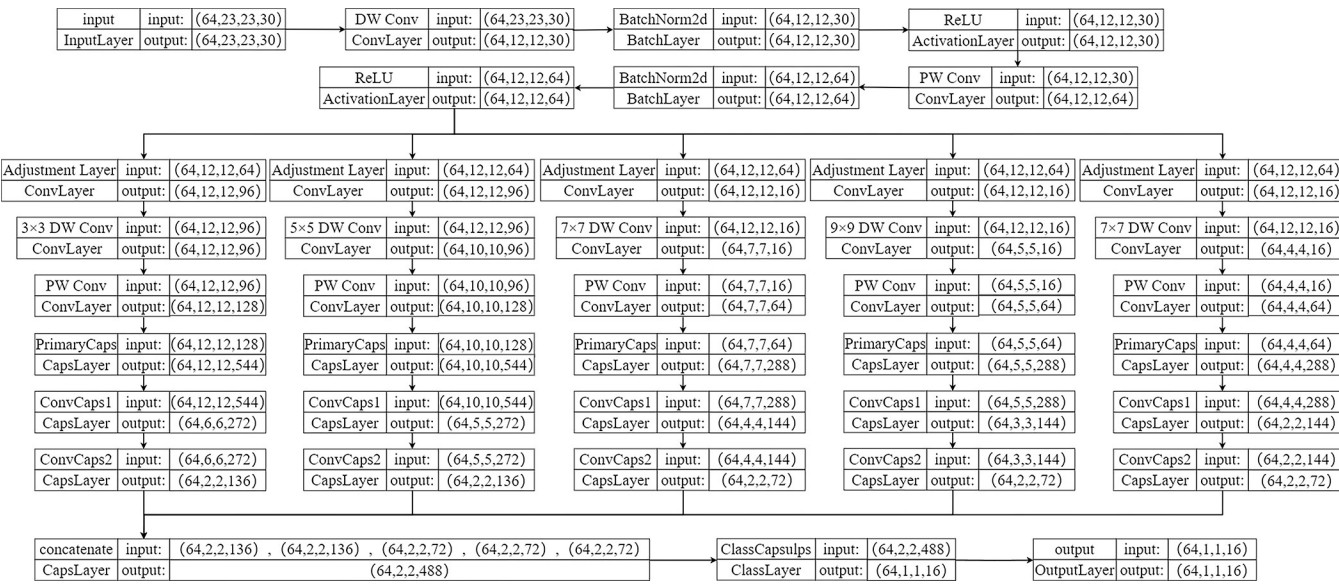

**Fig 7. The proposed MDSC-Net architecture.** The final layer of this architecture is specifically designed using the Salinas dataset.

The batch size is 64 for all cases. To achieve optimal model performance for each dataset, epochs are 400 for KSC Dataset, 300 for Pavia University and 200 for Salinas Dataset. The training was performed in a desktop computer running Windows 11, an Intel Core i7-12700H processor, an NVIDIA RTX 3070 Ti graphics card with 8GB of VRAM, and 16GB of system RAM.

To comprehensively evaluate the performance of the newly proposed MDSC-Net model, three available hyperspectral remote sensing imaging datasets were selected: the Kennedy Space Center Dataset, the Pavia University Dataset, and the Salinas Dataset. The details of the dataset are as follows:

1. The KSC Dataset was acquired from the area around the Kennedy Space Center, USA. It is made up of 13 categories across 224 spectral bands, reduced to 176 after removing water vapor noise. The resolution of the original image is 512×614 pixels, equating to an 18m×18m area per pixel. For computational efficiency in the MDSC-Net model, the band count was reduced to 20 through PCA and the images were resized to 532×634 pixels, further segmented into 21×21 pixel blocks. All background pixel blocks (labeled as zero) were excluded from the experimental dataset. The category details of the Kennedy Space Center dataset are shown in Table 1. In the entire dataset, there are 260 samples in the training set, 1823 samples in the validation set, and 3128 samples in the test set, with the training set comprising 5% of the total samples.

2. The Pavia University Dataset was collected by the University of Pavia, Italy, which is widely used in hyperspectral image classification and clustering algorithm research. The dataset is 610×340 pixels in size, has a spatial resolution of 1.3 meters, and initially contains 103 bands. It covers three different scenes: urban, rural, and wilderness, showcasing a variety of ground objects, including various buildings, land covers, and plant types. After compressing to 20 bands and dividing into 19×19 pixel blocks, the category details of the Pavia University dataset are shown in Table 2. In the entire dataset, there are 4277 samples in the training set, 11549 samples in the validation set, and 26950 samples in the test set, with the training set comprising 10% of the total samples.

**Table 1. Category details of the KSC dataset.**

| Index | Land Cover Type | Training Set Samples | Validation Set Samples | Test Set Samples | Total Samples |
|---|---|---|---|---|---|
| 0 | Scrub | 38 | 266 | 457 | 761 |
| 1 | Willow swamp | 12 | 85 | 146 | 243 |
| 2 | CP hammock | 13 | 90 | 153 | 256 |
| 3 | Slash pine | 13 | 88 | 151 | 252 |
| 4 | Oak/broadleaf | 8 | 56 | 97 | 161 |
| 5 | Hardwood | 11 | 80 | 138 | 229 |
| 6 | Swamp | 5 | 37 | 63 | 105 |
| 7 | Graminoid marsh | 22 | 151 | 258 | 431 |
| 8 | Spartina marsh | 26 | 182 | 312 | 520 |
| 9 | Cattail marsh | 20 | 141 | 243 | 404 |
| 10 | Salt marsh | 21 | 147 | 251 | 419 |
| 11 | Mud flats | 25 | 176 | 302 | 503 |
| 12 | Water | 46 | 324 | 557 | 927 |

3. Designed for hyperspectral image classification, the Salinas Dataset was obtained around the region of California's Salinas Valley, USA, by using the Airborne Visible/Infrared Imaging Spectrometer (AVIRIS). It features images at 512×217 pixels, a 3.7m spatial resolution, and originally 224 bands, reduced to 204 after noise removal. The ground truth consists of 16 types of features, including vegetables, bare soil, and vineyards. The original image is preprocessed to obtain blocks with 30 bands and a size of 23×23 pixels. The category details of the Salinas dataset are also shown in Table 3. In the entire dataset, there are 2706 samples in the training set, 25711 samples in the validation set, and 25712 samples in the test set, with the training set comprising 5% of the total samples.

## 4.2 Evaluation indicators

Different widely utilized quantitative measurement methods are utilized for assessing the network model proposed. These include Overall Accuracy (OA), which reflects the proportion of correctly classified samples, serving as an intuitive performance measure; Average Accuracy (AA), ensuring balanced performance assessment across different categories; Precision, measuring the accuracy of predicted positive samples; Recall, indicating the proportion of actual positives correctly predicted; the Kappa coefficient, evaluating the model's performance surpassing random levels; and the F1 Score (F1), providing a comprehensive assessment of classification accuracy and recall by calculating the harmonic mean of Precision and Recall. The

**Table 2. Category details of the Pavia University dataset.**

| Index | Land Cover Type | Training Set Samples | Validation Set Samples | Test Set Samples | Total Samples |
|---|---|---|---|---|---|
| 0 | Asphalt | 663 | 1989 | 3979 | 6631 |
| 1 | Meadow | 1865 | 5595 | 11189 | 18649 |
| 2 | Metal sheets | 135 | 406 | 812 | 1354 |
| 3 | Shadows | 95 | 284 | 568 | 947 |
| 4 | Gravel | 210 | 630 | 1259 | 2099 |
| 5 | Bitumen | 133 | 399 | 798 | 1330 |
| 6 | Bricks | 368 | 1105 | 2209 | 3682 |
| 7 | Bare soil | 503 | 1509 | 3017 | 5029 |
| 8 | Trees | 306 | 919 | 1839 | 3064 |

**Table 3. Category details of the Salinas dataset.**

| Index | Land Cover Type | Training Set Samples | Validation Set Samples | Test Set Samples | Total Samples |
|---|---|---|---|---|---|
| 0 | Brocoligreenweeds_1 | 101 | 704 | 1204 | 2009 |
| 1 | Brocoligreenweeds_2 | 186 | 1304 | 2236 | 3726 |
| 2 | Fallow | 99 | 692 | 1185 | 1976 |
| 3 | Fallowroughplow | 70 | 488 | 836 | 1394 |
| 4 | Fallow_smooth | 134 | 937 | 1607 | 2678 |
| 5 | Stubble | 198 | 1386 | 2375 | 3959 |
| 6 | Celery | 179 | 1253 | 2147 | 3579 |
| 7 | Grapes_untrained | 564 | 3945 | 6762 | 11271 |
| 8 | Soilvinyarddevelop | 310 | 2171 | 3722 | 6203 |
| 9 | Cornsenescedgreen_weeds | 164 | 1147 | 1967 | 3278 |
| 10 | Lettuceromaine4wk | 53 | 374 | 641 | 1068 |
| 11 | Lettuceromaine5wk | 96 | 674 | 1157 | 1927 |
| 12 | Lettuceromaine6wk | 46 | 321 | 549 | 916 |
| 13 | Lettuceromaine7wk | 54 | 375 | 641 | 1070 |
| 14 | Vinyard_untrained | 363 | 2544 | 4361 | 7268 |
| 15 | Vinyardverticaltrellis | 90 | 632 | 1085 | 1807 |

formulae for these metrics are as follows:

$$OA = \frac{1}{N} \sum_{i=1}^{r} x_{ii} \tag{7}$$

$$AA = \frac{1}{N} \sum_{i=1}^{r} Accuracy_i \tag{8}$$

$$Precision = \frac{TP}{TP + FP} \tag{9}$$

$$Recall = \frac{TP}{TP + FN} \tag{10}$$

$$Kappa = \frac{N \sum_{i=1}^{r} x_{ii} - \sum_{i=1}^{r} (Nx_{i.})}{N^2 - \sum_{i=1}^{r} (Nx_{i.})} \tag{11}$$

$$F_1 = \frac{2 * Precision * Recall}{Recall + Precision} \tag{12}$$

In the context of the equation, is the total number of categories; $N$ is the total number of samples; $x_{ii}$ represents the number of samples of class $i$ correctly identified as class $i$; $Accuracy_i$ is the precision of class $i$; True Positives ($TP$) denotes the number of actual positives correctly identified; False Positives ($FP$) indicates incorrect positive identifications among negatives; False Negatives ($FN$) represents missed positives, where actual positives are incorrectly classified as negatives.

## 4.3 Results of ablation studies

To validate the effectiveness of the proposed MDSC-Net model, ablation experiment were conducted on the Salinas dataset. The performance of different models was compared using three metrics: Overall Accuracy (OA), Average Accuracy (AA), and Kappa Score(Kappa), with consistent parameter settings and training strategies across models. Two comparative models have been selected: one is the capsule network model utilizing single-scale instead of multi-scale convolution kernel, and the other is a multi-scale convolutional network model using max-pooling layer directly instead of dynamic routing mechanism in the capsule network. Description of each model is shown in Table 4, with the comparative results of the ablation studies presented in Table 5.

Table 5 shows that the overall performance of MDSC-Net has been significantly improved after extracting multi-level features through multi-scale convolution kernel and replacing the max-pooling layer with the dynamic routing mechanism in capsule network. Specifically, compared with MDSC-0 (the capsule network model with single-scale convolutional kernel), MDSC-Net has improved the three key performance indexes of OA, AA and Kappa by 1.94%, 1.91% and 1.23% respectively. Against MDSC-1 (the multi-scale convolutional network model with max-pooling layer), the improvements were 7.44%, 7.50%, and 7.44% in the same metrics. The comparison between MDSC-0 and MDSC-Net highlights that the multi-scale convolutional kernel can capture rich multi-level features, thus effectively enhancing the capsule network's capability to process ground detail and multi-level information in HSI. The comparison between MDSC-1 and MDSC-Net indicates that substituting traditional max-pooling with the dynamic routing mechanism of capsule networks prevents the loss of precise location and posture information of objects, and realizes the translation invariance of the enhanced model features, and further improve the ability of feature extraction.

The classification results of ablation experiments are presented in Fig 8, including: (a) the original image of the dataset; (b) the manually annotated ground truth; (c) the output from the capsule network model using only single-scale convolutional kernel; (d) the output from the multi-scale convolutional network model with max-pooling layer; and (e) the output from the MDSC-Net model, which displayed more refined features compared to (c) and (d). This result is attributed to MDSC-Net's ability to extract rich hierarchical features through the application of multi-scale convolutional kernel, and effectively preserves object posture information via the capsule network, thereby more comprehensively extracting the feature details of HSI.

## 4.4 Results of comparative studies

To verify the effectiveness of the proposed MDSC-Net model, comparisons were made with existing models on the Kennedy Space Center dataset, the Pavia University dataset, and the Salinas dataset. The comparative models included SPP [28], DCNN [29], 3-D CNN [30], SPL-SR [31], AFLA-SCNN [32], and MLGSC [33]. Quantitative analysis was performed using the metrics Overall Accuracy (OA), Average Accuracy (AA), Recall, and F1 Score(F1).

The first comparative experiment, as shown in Fig 9, demonstrates the classification results of various classification models on the Kennedy Space Center dataset. Different subgraphs are

**Table 4. Combination structure of MDSC-Net.**

| Network Model | Description |
|---|---|
| MDSC-0 | Capsule network model utilizing only single-scale convolutional kernels |
| MDSC-1 | Multi-scale convolutional network model employing max-pooling layers |
| MDSC-Net | Multi-scale network model with an alternative to max-pooling layers |

Table 5. Results of ablation experiment.

| Network Model | OA(%) | AA(%) | Kappa(%) |
|---|---|---|---|
| MDSC-0 | 97.42 | 97.01 | 98.02 |
| MDSC-1 | 92.43 | 91.96 | 92.36 |
| MDSC-Net | 99.31 | 98.86 | 99.23 |

labeled in the Figure: (a) the RGB composite image; (b) the manually annotated ground truth; and (c) to (h) correspond to the outputs of several advanced classification techniques, including SPP, DCNN, 3-D CNN, SPL-SR, CNN_HSI, SpectralNET, and the proposed MDSC-Net model in (i).

Table 6 shows the quantitative analysis of the segmentation results on Kennedy Space Center dataset. The MDSC-Net model shows better classification accuracy than all other methods with AA, OA, FI, and Recall of 93%, 94%, 94%, and 92% respectively. The experiment shows that compared with SPP, DCNN, and 3-D CNN, the comparative metrics AA, OA, Recall, and F1 Score improved by 1% to 8% respectively. The overall parameter is higher than CNN_HSI and lower than SpectralNET. It can be concluded that the proposed MDSC-Net provides superior classification results on the Kennedy Space Center dataset with an acceptable parameter volume.

The second experiment, illustrated in Fig 10, presents the classification results of various models on the Pavia University dataset, with our proposed method achieving the most accurate segmentation results.

Table 7 lists the segmentation results on the Pavia University dataset. Compared to the six comparative models, the MDSC-Net has shown significant improvements in AA, OA, FI, and Recall by 97%, 98%, 94%, and 92% respectively. In summary, the MDCS-Net exhibited better segmentation outcomes than the benchmark algorithms on the Pavia University dataset. It showed superior performance with reasonable computational parameters, offering significant advantages over other models. These results confirm the robustness and broad applicability of MDSC-Net.

In the third experiment, as shown in Fig 11, the proposed method achieved better classification results on the Salinas dataset, with finer extraction results from minor to major categories, closely approximating the manually annotated Ground Truth.

As can be seen from Table 8, for Salinas dataset, the proposed method still maintained the most advanced performance, with AA, OA, F1 Score, and Recall all at 99%.

For hyperspectral remote sensing image classification, some models are complex [34–36], while others capture insufficient semantic information [37–39], necessitating efficient classification algorithms that ensure high accuracy while quickly processing large volumes of remote sensing image data. The introduction of multi-scale convolution strategy increases the computational complexity due to the parallel computation involving multiple scale convolution

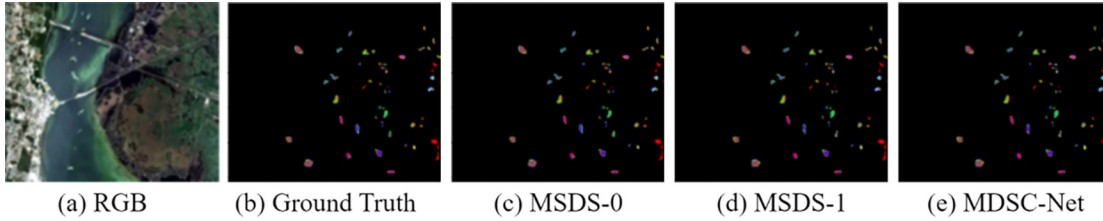

(a) RGB          (b) Ground Truth          (c) MSDS-0          (d) MSDS-1          (e) MDSC-Net

**Fig 8. Figure of intermediate results of ablation experiment.**

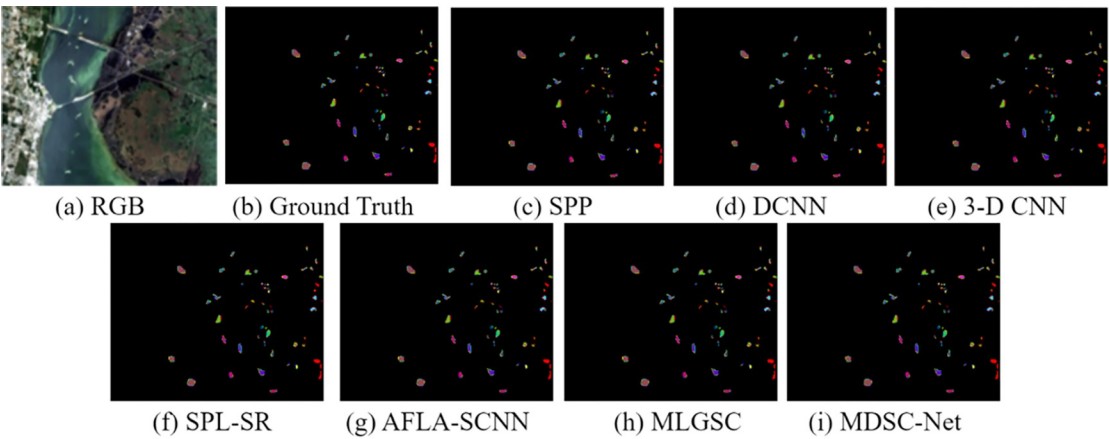

**Fig 9. Qualitative comparative experimental results on the KSC dataset.**

layers. To address this, the deep separable convolution technique is used to independently process the spatial feature extraction of each channel through two-dimensional convolution kernel, thereby reducing the number of parameters without sacrificing the overall model accuracy.

Traditional convolution operations handling spatial feature extraction and channel fusion as one unit lead to a significant parameter count (i.e., 64×k×k×d), while depthwise separable convolution partitions these processes. This separation significantly reduces the total number of parameters. Spatial feature extraction is conducted with parameters of k×k×d, while channel fusion is done with parameters of 64×1×1×d. As depicted in Fig 12 (parameter volume represented by blue bars and average accuracy by red lines), the proposed MDSC-Net demonstrates superior accuracy compared to the hierarchical multi-scale concatenation net (HMC-Net) by effectively utilizing depthwise separable convolution.

This approach not only enhances accuracy but also reduces the parameter count significantly, making it more suitable for real-world image classification tasks. This is attributed to MDSC-Net's adoption of multi-scale convolution kernels, depthwise separable convolutions, and capsule networks, which together enhance its ability to efficiently process and interpret complex image data while minimizing computational overhead.

## 4.5 Cross-validation experiment and analysis

To ensure the effectiveness and generalization ability of the proposed MDSC-Net model, this study employs 5-fold cross-validation on the Kennedy Space Center dataset. In the 5-fold cross-validation, the dataset is equally divided into five subsets, with each subset used as the test set in turn, while the remaining four subsets serve as the training set. This validation method effectively reduces random errors and the risk of overfitting during the model

**Table 6. Quantitative comparative experimental results on the KSC dataset.**

| Quantitative Metrics | SPP | DCNN | 3-D CNN | SPL-SR | AFLA-SCNN | MLGSC | MDSC-Net |
|---|---|---|---|---|---|---|---|
| AA(%) | 92 ± 0.1 | 92±0.3 | 86±0.4 | 91±0.3 | 91±0.02 | 88±0.01 | **92±0.1** |
| OA(%) | 91 ± 0.1 | 93±0.3 | 93±0.3 | 92±0.3 | 92±0.01 | 89±0.02 | **94±0.2** |
| F1(%) | 95 ± 0.1 | 94±0.03 | 86±0.4 | 92±0.3 | 92±0.03 | 89±0.02 | **93±0.3** |
| Recall(%) | 93 ± 0.1 | 0.93±0.3 | 93±0.3 | 93±0.3 | 91±0.01 | 87±0.07 | **94±0.2** |

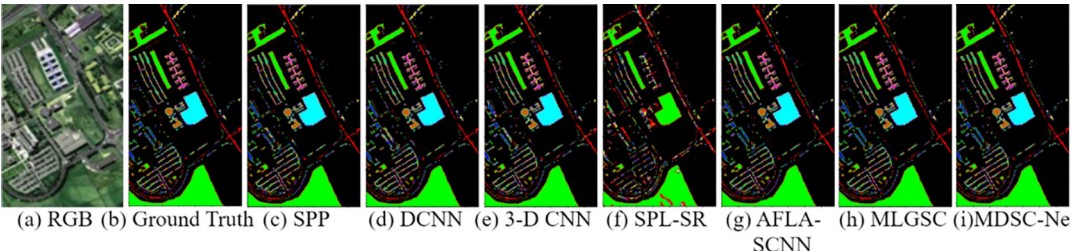

**Fig 10. Qualitative comparative experimental results on the Pavia University dataset.**

evaluation process, ensuring the reliability and consistency of the experimental results. The confusion matrix, averaged over the five test sets, is shown in the Fig 13.

From the comparative experiments, model parameter experiments, and cross-validation experiments, the following observations can be made:

By employing multi-scale convolutional kernels, depthwise separable convolutions, and capsule networks, MDSC-Net can effectively extract complex ground object information from hyperspectral images, maintain the translational invariance of features, and significantly reduce computational complexity and the number of parameters, thereby optimizing performance and efficiency.

Compared to state-of-the-art models, MDSC-Net performs better in terms of Overall Accuracy (OA), verifying its consistency and stability across all categories. Observing the confusion matrix, it is evident that MDSC-Net achieves high classification accuracy even in small sample categories. For instance, in categories 1, 4, 5, and 6 of the Kennedy Space Center dataset, despite having fewer training samples, the classification accuracy does not significantly decrease. However, the MDSC-Net model has certain limitations. Due to the reduction in the number of parameters, the time complexity is relatively increased, requiring approximately 4 hours of training on a computer equipped with a 3070 Ti GPU.

## 5 Conclusion

This article presents a Multi-Scale Depthwise Separable Capsule Network for hyperspectral image (HSI) classification, demonstrating significant effectiveness in the classification of hyperspectral imagery. By employing multi-scale convolutional kernels, the network is capable of capturing features across different scales, effectively deciphering the complex terrain details and hierarchical information in HSIs. Moreover, the model employs depthwise separable convolutions to streamline processing, achieving efficient feature extraction with reduced computational load. The incorporation of capsule networks, as an improvement over traditional max-pooling layers, allows the model to reduce the size of feature maps while maintaining translational invariance and preventing the loss of terrain posture information. Ablation studies show that the MDSC-Net, with its multi-scale convolutional kernels and capsule networks, significantly enhances HSI classification performance. Comparative experiments on three HSI

**Table 7. Quantitative comparative experimental results on the Pavia University dataset.**

| Quantitative Metrics | SPP | DCNN | 3-D CNN | SPL-SR | AFLA-SCNN | MLGSC | MDSC-Net |
|---|---|---|---|---|---|---|---|
| AA(%) | 95 ± 0.3 | 88±0.4 | 94±0.3 | 88±0.4 | 94±0.03 | 94±0.04 | **97±0.2** |
| OA(%) | 93 ± 0.4 | 92±0.3 | 94±0.3 | 84±0.4 | 93±0.03 | 95±0.03 | **98±0.1** |
| F1(%) | 94± 0.3 | 89±0.2 | 92±0.1 | 86±0.2 | 95±0.03 | 94±0.02 | **98±0.1** |
| Recall(%) | 96 ± 0.2 | 90±0.3 | 94±0.3 | 89±0.4 | 95±0.03 | 95±0.03 | **97±0.2** |

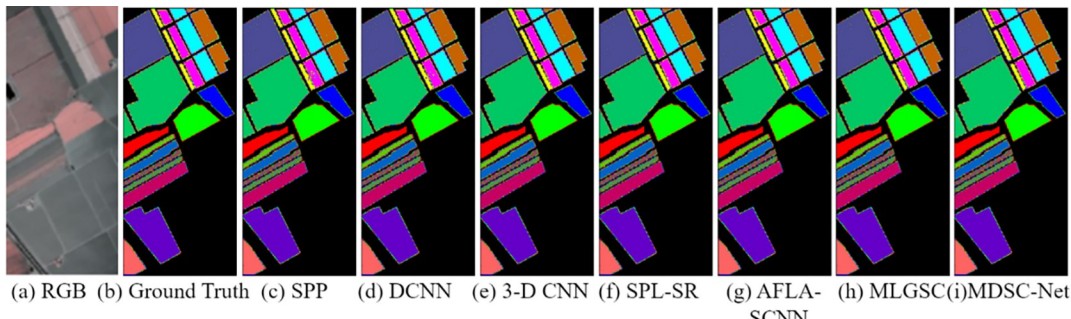

**Fig 11. Qualitative comparative experimental results on the Salinas dataset.**

datasets further affirm the performance advantages of this model over existing ones, including structural simplicity, lower computational complexity, and higher classification accuracy.

For future work, this model can be utilized in precision agriculture for crop health monitoring. By analyzing hyperspectral images, issues such as pest infestations and nutrient

**Table 8. Quantitative comparative experimental results on the Salinas dataset.**

| Quantitative Metrics | SPP | DCNN | 3-D CNN | SPL-SR | AFLA-SCNN | MLGSC | MDSC-Net |
|---|---|---|---|---|---|---|---|
| AA(%) | 81±0.3 | 93±0.3 | 98±0.2 | 96±0.2 | 98±0.02 | 97±0.03 | **99±0.1** |
| OA(%) | 76±0.4 | 89±0.3 | 96±0.2 | 93±0.2 | 97±0.02 | 98±0.03 | **99±0.4** |
| F1(%) | 83±0.2 | 92±0.5 | 97±0.3 | 95±0.2 | 98±0.02 | 97±0.02 | **99±0.2** |
| Recall(%) | 78±0.3 | 87±0.3 | 99±0.2 | 94±0.3 | 97±0.03 | 99±0.04 | **99±0.1** |

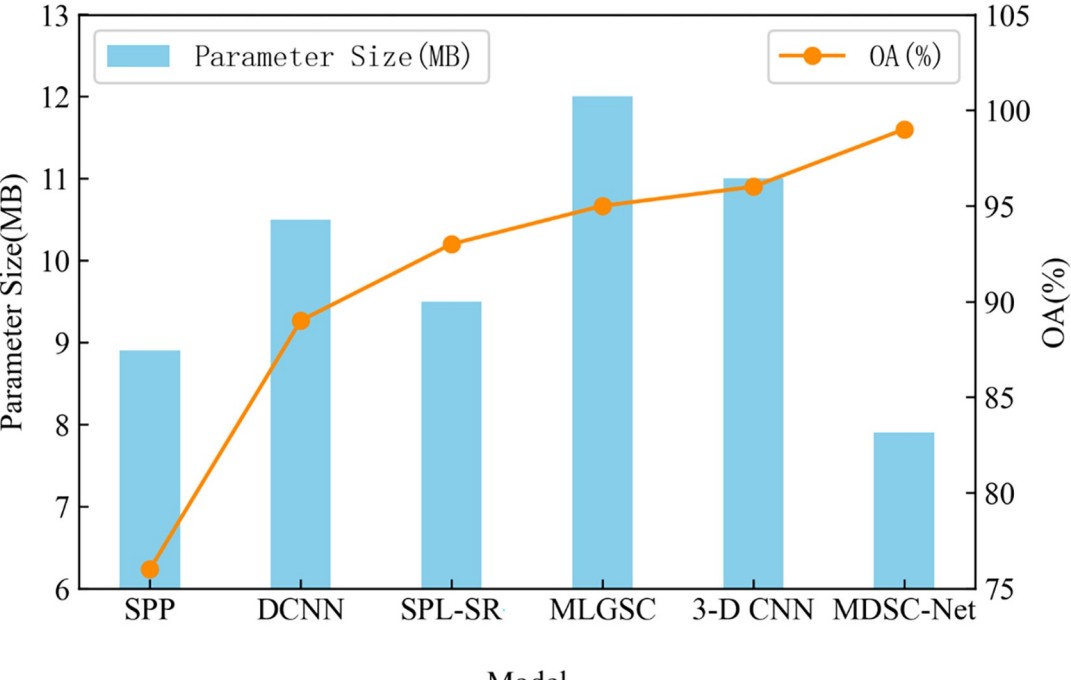

**Fig 12. Comparison chart of model parameter quantity and OA.**

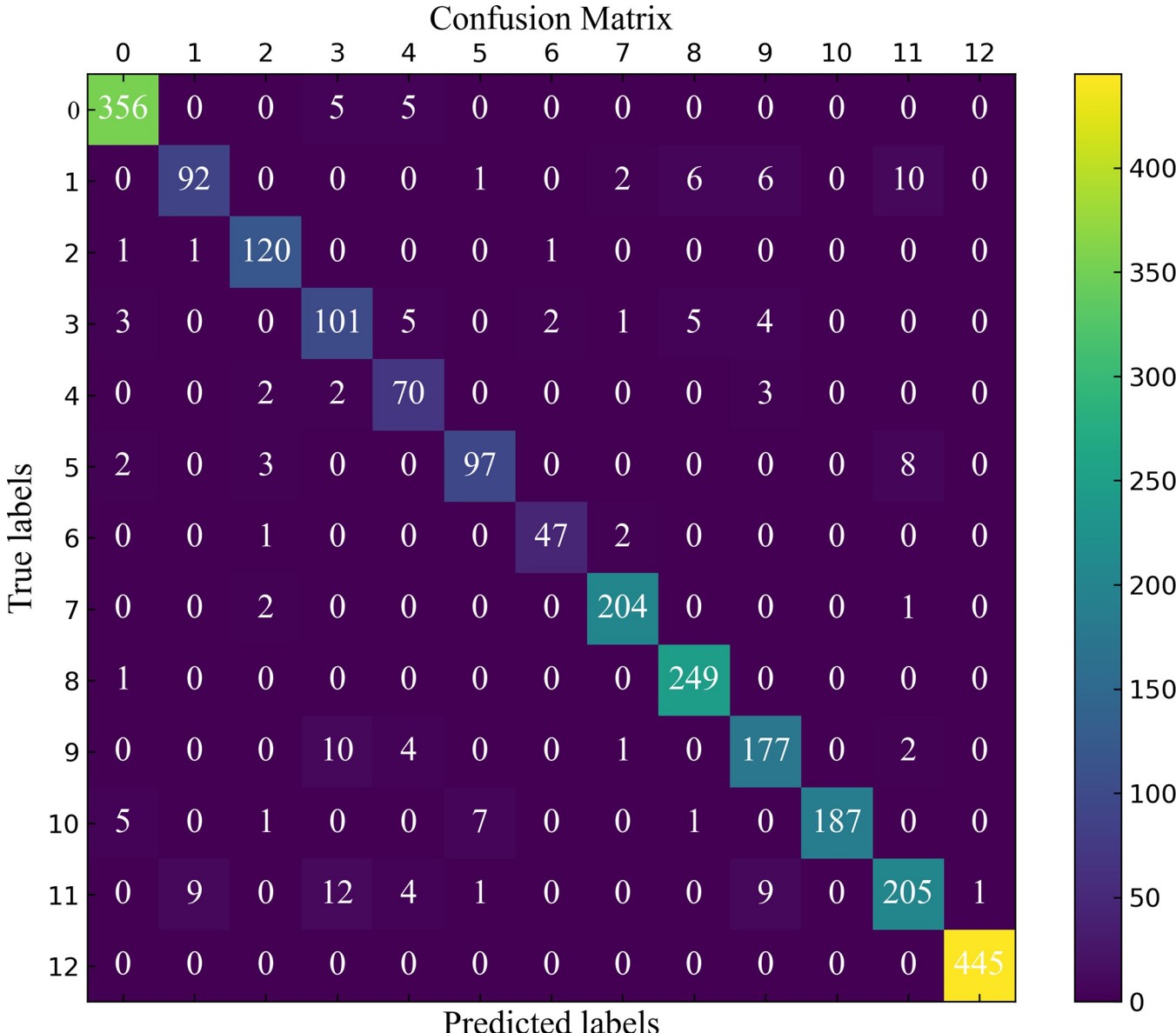

**Fig 13. Confusion matrix.**

deficiencies can be identified, providing precise field management recommendations. Additionally, this model can have significant applications in urban planning. It can be used for land use classification and change detection, assisting planners in accurately assessing urban expansion and the dynamic changes in land use.

This study has some limitations, such as the potential for improved classification accuracy on datasets with a smaller number of samples. Additionally, the generalizability of the model to other hyperspectral image datasets remains to be verified. Challenges also exist regarding processing speed in real-time applications and computational demands on resource-limited devices like embedded systems. Future research should focus on exploring effective ways to handle large datasets under constrained computational resources. Incremental Principal Component Analysis (IPCA) and incremental learning methods can be employed to gradually

extract the main components of the data, reducing the demand for computational resources and storage, and supporting dynamic model updates to lower computational costs. Moreover, the MDSC-Net model faces the challenge of maintaining efficient performance on battery-powered devices. This requires adjustments through quantization and sparsity techniques to reduce energy consumption and the exploration of adaptive computing methods to achieve an optimal balance between performance and energy consumption. Furthermore, investigating cross-modal learning strategies, such as combining HSI data with LiDAR data, may enhance the accuracy and robustness of HSI classification, thereby improving the performance and applicability of the MDSC-Net model in real-world scenarios. Finally, to enhance the model's generalizability across different datasets, it is crucial to test and optimize it on a more diverse range of datasets.

## Supporting information

**S1 File.**
(ZIP)

## Author Contributions

**Conceptualization:** Lin Wei, Haoxiang Ran, Yuping Yin, Huihan Yang.

**Data curation:** Lin Wei, Haoxiang Ran, Yuping Yin, Huihan Yang.

**Formal analysis:** Lin Wei, Haoxiang Ran, Yuping Yin.

**Funding acquisition:** Lin Wei, Haoxiang Ran, Yuping Yin.

**Investigation:** Lin Wei, Haoxiang Ran, Yuping Yin.

**Methodology:** Lin Wei, Haoxiang Ran, Yuping Yin.

**Project administration:** Lin Wei, Haoxiang Ran, Yuping Yin.

**Resources:** Lin Wei, Haoxiang Ran, Yuping Yin.

**Software:** Haoxiang Ran.

**Supervision:** Lin Wei, Haoxiang Ran, Yuping Yin.

**Validation:** Lin Wei, Haoxiang Ran, Yuping Yin, Huihan Yang.

**Visualization:** Lin Wei, Haoxiang Ran, Huihan Yang.

**Writing – original draft:** Haoxiang Ran, Huihan Yang.

**Writing – review & editing:** Lin Wei, Haoxiang Ran, Yuping Yin, Huihan Yang.

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
