## [Decision Letter · Decision Letter 0]

7 Jun 2024

PONE-D-24-15298Multi-Scale Depthwise Separable Capsule Network for Hyperspectral Image ClassificationPLOS ONE

Dear Dr. HaoXiang,

Thank you for submitting your manuscript to PLOS ONE. After careful consideration, we feel that it has merit but does not fully meet PLOS ONE’s publication criteria as it currently stands. Therefore, we invite you to submit a revised version of the manuscript that addresses the points raised during the review process.

We look forward to receiving your revised manuscript.

Kind regards,

Narendra Khatri, Ph.D.

Academic Editor

PLOS ONE

Journal Requirements:

"This work was supported in part by the Basic Scientific Research Projects of Education Department of Liaoning Province under Grant LJKMZ20220687, "Hyperspectral Image Classification Based on Multi-layer Deep Learning Method with Small Samples". It was also supported by the Liaoning Provincial Natural Science Foundation under Grant 1704681991881, "Research on High-precision Classification of Hyperspectral Images Using Lightweight Feature Fusion Based on Self-organizing Multi-layer Heterogeneous Deep Learning Method". Furthermore, the research received funding from the Science and Technology Plan Project of Huludao City under Grant 2023JH (1) 4/04b, "Research on High-precision Classification of Hyperspectral Images Based on multi-layer attention and puzzle networks"

5. We notice that your supplementary figures are uploaded with the file type 'Figure'. Please amend the file type to 'Supporting Information'. Please ensure that each Supporting Information file has a legend listed in the manuscript after the references list.

6. We notice that your supplementary tables are included in the manuscript file. Please remove them and upload them with the file type 'Supporting Information'. Please ensure that each Supporting Information file has a legend listed in the manuscript after the references list.

**Additional Editor Comments:**

Major Revision

Reviewers' comments:

Reviewer's Responses to Questions

**Comments to the Author**

1. Is the manuscript technically sound, and do the data support the conclusions?

Reviewer #1: Yes

Reviewer #2: Yes

2. Has the statistical analysis been performed appropriately and rigorously? 

Reviewer #1: Yes

Reviewer #2: Yes

3. Have the authors made all data underlying the findings in their manuscript fully available?

Reviewer #1: No

Reviewer #2: Yes

4. Is the manuscript presented in an intelligible fashion and written in standard English?

Reviewer #1: No

Reviewer #2: Yes

5. Review Comments to the Author

Reviewer #1: 1.The research paper lacks gaps of stud as papers cited are form year 2023 and not included from 2024

2.The graphical representation of analysis of the parameters are not done properly

3. Differentiate between equation number 4 and 5 i.e which parameters they are representing

4. The paper is not set according to the format of the journal

5.Flow of the paper is not streamlined

Reviewer #2: The author(s) have presented an intriguing approach to improving hyperspectral image classification by using a Multi-Scale Depthwise Separable Capsule Network. While the manuscript is well-structured and the topic is highly relevant, I have a few comments and suggestions that I believe could further enhance the clarity and impact of your work.:

Comments to the Author(s)

1. The introduction is well-written but could benefit from a clearer statement of the contributions. Highlight the novelty and significance of your approach explicitly.

2. The literature review is comprehensive but ensure it includes the most recent works in hyperspectral image classification. Discuss recent trends and how your work fits into these trends.

3. The experimental setup is well-documented, but more details about the datasets are needed. Describe characteristics like the number of samples, classes, and resolution to provide context for the results.

4. Clarify the data split between training and testing, and mention whether cross-validation was performed.

5. The proposed Multi-Scale Depthwise Separable Capsule Network (MDSC-Net) is detailed. Include a diagram to illustrate the network architecture for better understanding.

6. Ensure all mathematical notations and equations are clearly defined and explained. Additional explanations or examples might be helpful.

7. The results are promising but need a more detailed comparison with state-of-the-art methods. Highlight specific areas where your method outperforms others and discuss any potential limitations.

8. Include more visualizations such as confusion matrices and ROC curves to complement the quantitative results.

9. The future work section should provide more concrete directions for further research. Discuss potential applications of the proposed method and any foreseeable challenges.

10. Address any limitations of the current study and suggest how these could be overcome in future research.

These comments aim to enhance the paper's clarity, completeness, and impact, ensuring it meets the high standards expected by the journal.

6. PLOS authors have the option to publish the peer review history of their article (what does this mean?). If published, this will include your full peer review and any attached files.

Reviewer #1: **Yes: **Dr. Alankrita Aggarwal Department of Computer Science and Engineering Apex Institute of Technology University Institute of Sciences Chandigarh University Mohali 140413 Punjab India

Reviewer #2: No

---

## [Author Response · Author response to Decision Letter 0]

16 Jul 2024

Dear editor,

Please find enclosed our revised manuscript entitled "Multi-Scale Depthwise Separable Capsule Network for Hyperspectral Image Classification" (Manuscript ID: PONE-D-24-15298). We have carefully addressed all the points raised by the reviewers and the academic editor.

Reviewer #1：

1. The research paper lacks gaps of stud as papers cited are form year 2023 and not included from 2024.

Response: In the introduction, we have included an exposition of recent development trends, incorporating some of the latest literature from 2024. These references cover the recent applications of Convolutional Neural Networks (CNNs) in hyperspectral image (HSI) classification tasks, as well as the emerging trends in the application of capsule networks for HSI classification.

2. The graphical representation of analysis of the parameters are not done properly.

Response: We have reconstructed the parameter analysis charts, adjusted the model sequence, and optimized the font. Additionally, the line style of the axes has been modified to face inward. Two of the earlier models have been replaced with two of the latest models, and the corresponding data have been updated.

3. Differentiate between equation number 4 and 5 i.e which parameters they are representing.

Response: I previously mistakenly wrote "OA" as "AA" in Equation 5. This error has now been corrected, and other equations have been thoroughly checked to ensure no similar mistakes are present.

4. The paper is not set according to the format of the journal.

Response: The entire document has been thoroughly reviewed for formatting, and revisions have been made to sections that did not meet the journal's requirements. Specific modifications include the layout of tables, the formatting of equations and figures, as well as correcting format issues in some references.

5. Flow of the paper is not streamlined.

Response: Revisions have been made to the sections of the paper where the flow was unreasonable, and additional content has been included to enhance the overall coherence of the paper. Specifically, the description in the introduction, the explanation of the EM algorithm in Chapter 2, and the arrangement of the sections in the experimental part have been adjusted and optimized.

Reviewer #2：

1. The introduction is well-written but could benefit from a clearer statement of the contributions. Highlight the novelty and significance of your approach explicitly.

Response: The introduction section has been revised to clearly articulate the contributions of this paper using more concise language. Descriptions of the contributions have been added, and previously unreasonable parts of the narrative sequence have been adjusted.

2. The literature review is comprehensive but ensure it includes the most recent works in hyperspectral image classification. Discuss recent trends and how your work fits into these trends.

Response: In the introduction, we have included a discussion on recent development trends in hyperspectral image (HSI) classification, covering the latest applications of Convolutional Neural Networks (CNNs) in HSI classification tasks and the emerging trends in the application of capsule networks for HSI classification. Additionally, it has been highlighted that this study also employs capsule network-related techniques for HSI classification tasks.

3. The experimental setup is well-documented, but more details about the datasets are needed. Describe characteristics like the number of samples, classes, and resolution to provide context for the results.

Response: In Section 4.1, we have added three tables to introduce the datasets, detailing the class names and the number of samples for each dataset, as well as the division of the training, validation, and test sets. Additionally, we have provided more detailed information about the datasets, including resolution and the proportion of the training set.

4. Clarify the data split between training and testing, and mention whether cross-validation was performed.

Response: In Subsection 4.5, we have added a discussion on cross-validation, providing a detailed explanation of the five-fold cross-validation method, including the procedure of conducting five-fold cross-validation and the resulting outcomes.

5. The proposed Multi-Scale Depthwise Separable Capsule Network (MDSC-Net) is detailed. Include a diagram to illustrate the network architecture for better understanding.

Response: We added a description and an illustration in Section 3.6 to depict the input and output dimensions of each network layer, along with details on layer types, names, kernel sizes, and quantities. This enhancement aims to provide a clearer understanding of the architecture of our proposed Multi-Scale Depthwise Separable Capsule Network (MDSC-Net).

6. Ensure all mathematical notations and equations are clearly defined and explained. Additional explanations or examples might be helpful.

Response: Upon review, we found that the explanation of the EM algorithm was overly simplified. Therefore, we have added definitions and explanations of Equations 1 and 2 in Chapter 2, along with an example to further elucidate the operational logic of the EM algorithm.

7. The results are promising but need a more detailed comparison with state-of-the-art methods. Highlight specific areas where your method outperforms others and discuss any potential limitations.

Response: In Section 4.4, we have added further comparative explanations of various advanced methods, replacing two earlier models with two latest ones and updating the corresponding data. In Section 4.5, we have included additional analysis, providing a detailed discussion on the aspects where our method outperforms these advanced models, as well as areas where it falls short.

8. Include more visualizations such as confusion matrices and ROC curves to complement the quantitative results.

Response: In Section 4.5, we have added the confusion matrix for the Kennedy Space Center dataset. This confusion matrix marks correctly and incorrectly classified data with varying shades of color to intuitively display the classification performance.

9. The future work section should provide more concrete directions for further research. Discuss potential applications of the proposed method and any foreseeable challenges.

Response: In the conclusion section, we have added a discussion on specific directions for future work. We analyzed two potential application scenarios for our proposed method and provided a detailed discussion on various challenges that may be encountered in three specific aspects.

10. Address any limitations of the current study and suggest how these could be overcome in future research.

Response: In Section 4.5, we compared our method with other advanced models and highlighted areas where our research falls short compared to these models. In the conclusion section, we discussed the potential limitations of the current research from multiple perspectives and provided possible solutions for these issues.

We have ensured that our manuscript now meets PLOS ONE's style requirements and have made necessary amendments as requested, including updating the financial disclosure and grant information, as well as correctly categorizing the supplementary files.We deeply appreciate the constructive feedback provided, which has helped us improve the quality of our manuscript. We look forward to receiving your further comments.If you have any queries, please don’t hesitate to contact me at the address below.

Sincerely,

Haoxiang RAN

Electronic and Information Engineering

Liaoning Technical University

188 Longwan South Street, Huludao, Liaoning, China

E-mail:3295906115@qq.com

July 16, 2024

---

## [Decision Letter · Decision Letter 1]

31 Jul 2024

Multi-Scale Depthwise Separable Capsule Network for Hyperspectral Image Classification

PONE-D-24-15298R1

Dear Dr. HaoXiang,

We’re pleased to inform you that your manuscript has been judged scientifically suitable for publication and will be formally accepted for publication once it meets all outstanding technical requirements.

Kind regards,

Narendra Khatri, Ph.D.

Academic Editor

PLOS ONE

Additional Editor Comments (optional):

Accept

Reviewers' comments:

Reviewer's Responses to Questions

**Comments to the Author**

1. If the authors have adequately addressed your comments raised in a previous round of review and you feel that this manuscript is now acceptable for publication, you may indicate that here to bypass the “Comments to the Author” section, enter your conflict of interest statement in the “Confidential to Editor” section, and submit your "Accept" recommendation.

Reviewer #1: All comments have been addressed

Reviewer #2: All comments have been addressed

2. Is the manuscript technically sound, and do the data support the conclusions?

Reviewer #1: Yes

Reviewer #2: (No Response)

3. Has the statistical analysis been performed appropriately and rigorously? 

Reviewer #1: Yes

Reviewer #2: (No Response)

4. Have the authors made all data underlying the findings in their manuscript fully available?

Reviewer #1: Yes

Reviewer #2: (No Response)

5. Is the manuscript presented in an intelligible fashion and written in standard English?

Reviewer #1: Yes

Reviewer #2: (No Response)

6. Review Comments to the Author

Reviewer #1: (No Response)

Reviewer #2: (No Response)

---

## [Editor Report · Acceptance letter]

16 Aug 2024

PONE-D-24-15298R1 

PLOS ONE

Dear Dr. Ran, 

I'm pleased to inform you that your manuscript has been deemed suitable for publication in PLOS ONE. Congratulations! Your manuscript is now being handed over to our production team.

Kind regards, 

on behalf of

Dr. Narendra Khatri 

Academic Editor

PLOS ONE